# Age, health and other factors associated with return to work for those engaging with a welfare-to-work initiative: a cohort study of administrative data from the UK's Work Programme

Judith Brown,[1] Srinivasa Vittal Katikireddi,[2] Alastair H Leyland,[2] Ronald W McQuaid,[3] John Frank,[4] Ewan B Macdonald[1]

For numbered affiliations see end of article.

**Correspondence to**
Dr Judith Brown;
Judith.Brown@glasgow.ac.uk

## ABSTRACT

**Objectives** To investigate the role of individual factors (including age, health and personal circumstances) and external factors associated with clients having a job start while engaging with the Work Programme and variations by benefit type.

**Setting** The UK Government's main return to work initiative (The Work Programme) in Scotland.

**Design** Piecewise Poisson regression to calculate incident rate ratios using administrative data from 2013 to 2016 to identify factors associated with job start.

**Participants** 4322 Employment and Support Allowance (ESA) clients not in work due to poor health and 8996 Jobseeker's Allowance (JSA) clients, aged 18–64 years, referred to the Work Programme between April 2013 and July 2014.

**Main outcome measures** Starting a job and the time to first job start after entering the Work Programme.

**Results** JSA clients (62%) were more likely to return to work (RTW) than ESA clients (20%). There is a strong negative relationship between age and the predicted probability of having a job start during the 2-year engagement with the programme for both JSA and ESA clients. JSA clients were most likely to RTW in the first 3 months, while for ESA clients the predicted probability of having a first job start was fairly constant over the 2 years. Health, including the number of health conditions, length of unemployment, client perception of job start and other individual factors were associated with job starts for both groups.

**Conclusions** Age plays an important role in influencing RTW; however, important potentially modifiable factors include the length of unemployment, the management of multimorbidity and the individual's perception of the likelihood of job start. Future welfare-to-work programmes may be improved by providing age-specific interventions which focus on health and biopsychosocial factors to enable more people to realise the potential health benefits of RTW.

## INTRODUCTION

Labour market participation is an important determinant of health and health inequalities, with efforts to increase paid employment

### Strengths and limitations of this study

► This is the first study to explore the role of individual age, health and other factors in returning to work, using large-scale administrative data collected over a client's 2 years participation in the Work Programme.

► The use of administrative data (rather than survey data as in previous evaluations) limited loss in follow-up; however, the analysis relies on routine operational data with limited health diagnostic and severity information available.

► Jobseeker's Allowance (JSA) and Employment Support Allowance (ESA) clients were analysed separately showing important differences in return to work between the two client groups (by definition those with and without an illness, health condition or disability that makes it difficult to work, although many JSA clients disclosed health conditions).

► The number of (self-disclosed) health conditions for JSA and ESA provides new insights.

► This study had modelled age as a continuous variable (1406 JSA clients and 1322 ESA clients aged 50 years and over) to better understand the relationship between age and RTW rather than the single age category (50+ years) of other studies.

thought by policymakers to improve health.[1–5] However, health status also inhibits returning to work from being unemployed, especially for those with multiple health problems.[6–9] In addition, age is closely connected to both health status and other difficulties in returning to work, with employment rates for the working age population declining sharply from over 80% of those aged 50 years in the UK, to around 60% of those aged 60 years with a steeper decline at older ages.[10] This paper uses a unique database to analyse the likelihood of unemployed people on the government's Work Programme (WP) returning to work. Specifically it considers the

role of age when comparing those with an illness, health condition or disability that makes it difficult to work (those receiving the Employment and Support Allowance (ESA)) and others on the programme (those receiving Jobseeker's Allowance (JSA)).

Due to the ageing population, older workers increasingly make up a greater proportion of the workforce in most economically developed countries.[11] Thus, recruiting and retaining older workers, and encouraging and enabling more people to work for longer, is a policy priority for governments and employers.[11 12] For instance, from 2014 to 2024 the UK will have 200000 fewer people aged 16–49 years and 3.2million more people aged 50 years to state pension age, but the latter with a lower average employment rate.[13]

Older workers face significant barriers in the labour market and are less likely to regain employment after job loss and are at increased risk of chronic health conditions, which contribute to job loss and may make re-employment difficult.[14–17] In addition, this age group may encounter barriers and factors that interact with their health including direct and indirect age discrimination,[18–20] skills gaps (especially in IT),[19] caring responsibilities, for example, for grandchildren or elderly parents[21] and the lack of flexible working opportunities.[22] Some individuals may experience multiple interacting or overlapping disadvantages, which may be difficult to resolve in isolation.[8 23] For older workers (in this study refers to those aged 50–64 years), there are added difficulties of separating the impact of biological ageing from the impacts of unemployment and health selection effects (eg, ill health leading to early voluntary retirement).[5]

The WP was the UK Government's flagship welfare-to-work initiative to help those more detached from the labour market to enter employment and reduce the time people spent on benefits. The design of the WP has parallels with other Organisation for Economic Cooperation and Development countries' active labour market policies for those on welfare or unemployed, in terms of moves towards delivery of general and specialist employment services through networks of private and not-for-profit organisations, usually through employment outcomes-based performance contracts, with a variety of forms of procurement.[24 25] In addition to the WP, outsourcing included services for the disabled in countries such as Australia and the Netherlands (mainly to not-for-profit or private organisations), Sweden, Denmark and the USA.[26–30]

The WP was launched throughout Great Britain in June 2011 as part of a sweeping programme of welfare reforms with final referrals in March 2017.[31] It required more people to either seek work or to undertake some form of work-related activity as a condition of receiving benefit.[32] The 2-year programme was delivered by a range of private, public and voluntary sector organisations across 18 regions. The Department for Work and Pensions (DWP) paid two or more prime contractors to provide directly or indirectly (through subcontractors) support to unemployed job seekers in each region (delivered as a 'black box' approach where much of the control over services provided was wielded by the contractor). One innovatory feature was that payments were by results with most payments being after a participant had sustained employment (although not necessary with the same employer) for a minimum time, rather than payments being mainly for job seekers participating or entering employment.

The unemployed including those out of work due to health reasons were required to participate in the WP, and others were able to volunteer to use the services depending on their circumstances. The WP supported two main groups of benefit claimants: JSA clients and ESA clients. JSA is a benefit for people who are unemployed but capable of work, usually paid to unemployed people if all of the following apply: they are aged 18 years to state pension age, not in full-time education, living in Great Britain, available for work, actively seeking work and work on average <16hours per week.[33] Those aged 18–24 years were a priority group and had to have been unemployed a shorter period (usually 6 months) before entering the WP than most older groups. ESA is a benefit for people who have an illness, health condition or disability that makes it difficult to work and requires participants to undergo a Work Capability Assessment.[34] Claimants may get ESA if their illness or disability affects their ability to work and if they are: under state pension age, not getting statutory sick pay or statutory maternity pay and have not gone back to work and are not getting JSA. ESA clients are not separated by age and often people become disabled or experience worsening health, and so join ESA, later in life. So generally ESA clients in our study are older than JSA clients.

One of the most extensive reports from the national WP evaluation to investigate factors influencing return to work (RTW) is a telephone survey of 4700 clients and a follow-up survey of 1800 of the same clients.[8] It found that after 2years on the programme, 67% of people were still not in work, and were more likely to be male, older than 55 years, have health conditions, few qualifications and no recent work experience. Because having a health condition or disability was shown to be an important factor we believe there is a need to investigate ESA clients separately since people with long-term health conditions and disabilities experience disproportionately lower employment rates. Data from 2015 show a significant gap in the employment rates of disabled (48%) and non-disabled people (80%),[35] and while the UK Government is committed to seeing 1 million more people in work over the next 10 years,[36] if there is to be any chance of achieving this, then there must be a focus on improving outcomes for the over 50s and for those with disabilities in these new schemes. Furthermore, the evaluation investigated clients by age categories (18–24, 25–34, 35–44, 45–54, 55–59 and 60+ years) and for some analyses grouped clients aged 50+ years into one group. The National Audit report of the WP did investigate JSA clients (termed easier-to-help) and ESA clients (termed harder-to-help) separately but they only reported on performance in getting people into work and not on the factors associated with RTW.[32]

Age UK reports divided over 50s into three age categories (50–54, 55–59 and 60+ years) and suggested that low job performance is not caused by a higher incidence of disability or health conditions but rather age itself was the main barrier to work.[37 38]

In order to better understand all clients aged 50 years and above, we investigated individual factors, personal circumstances and external factors associated with RTW of JSA and ESA clients separately, treating age as a continuous variable. The aim of this study was to answer the following research questions:

1. What is the relationship between age and returning to work for JSA and ESA clients engaged with the WP?
2. How does the likelihood of returning to work change over the period of their participation in the WP for JSA and ESA clients?
3. What other factors, including health, are associated with RTW for JSA and ESA clients?

## METHODS
### Description of cohort
'Supporting Older People into Employment' (SOPIE) is a mixed methods longitudinal study involving a collaboration between academics, a major WP provider (Ingeus) and the UK DWP. Full details on the study, including sample size, can be found in the protocol paper.[39] The study population was all clients who entered the Ingeus WP in Scotland between 1 April 2013 and 31 July 2014 (14 265 clients). After data cleaning the SOPIE cohort totalled 13 318 clients. The 947 clients were removed from the study population as they had significant missing baseline data including age (n=693); it was not possible to generate datazone from postcode (n=100); were aged 16–17 years (n=7); had lengths of unemployment which were not possible, for example, a client aged 20 years with 10 years of unemployment history (n=147). The cohort was followed up longitudinally for the 2 years they engaged with Ingeus on the WP.

### Variables
After referral to the WP, clients completed a baseline face-to-face assessment with an employment advisor. The individual factors collected in the baseline assessment and used in this study were: age, gender, length of unemployment prior to the WP, highest qualification, ethnicity, whether the client had health concerns which they believed would affect their ability to work, number of health conditions disclosed to advisor, client perception of their likelihood of starting a job and personal circumstances (caring responsibility other than children, housing status, parental status). Ethnicity was recoded as white British and all other due to sample size. The number of health conditions disclosed by clients were categorised into 0, 1, 2 and 3 or more for JSA clients and 0/1, 2, 3, 4 and 5 or more for ESA clients. As expected, ESA clients disclosed a greater number of health conditions. One hundred twenty-one (2.8%) ESA clients disclosed no

health conditions. This may have been a coding error or the client may have decided not to disclose their health condition to their advisor; hence, we coded 0 and 1 disclosed health conditions together (table 1).

Analyses of external factors were conducted using datazones. The 6976 datazones in Scotland have populations of between 500 and 1000 household residents,[40] and Ingeus determined the datazone from the client's postcode. For each client the research team added the 2016 Scottish Index of Multiple Deprivation (SIMD) quintile and the Scottish Government sixfold urban rural classification (large urban areas, other urban areas, accessible small towns, remote small towns, accessible rural, remote rural).[41 42] The SIMD ranks datazones from the most (number 1) to the least deprived.[41]

### Outcome measure
The primary outcome measures were the client starting a job and the length of time from beginning the WP to their first job start (months).

### Statistical analysis
To address the three research questions a mixture of descriptive statistics and regression analyses were used. All analyses were stratified by benefit type (JSA and ESA clients) given the large differences in RTW between the two groups. Counts and percentages were used to summarise categorical variables. The associations between benefit type and all the study variables were analysed using $X^2$ tests. Cox's proportional hazards models were initially used to determine the HRs of clients returning to work but the proportional hazards assumption was not met. We therefore approximated the survival model using a piecewise Poisson regression model—equivalent to a Cox model with baseline hazard able to vary between sections.[43] Split times used in the models were as follows: 0–3, 3–6, 6–9, 9–12 and 12–24 months (due to sample size a 3-month average probability is shown for the 12–24 months time period). We modelled age as a continuous variable using fractional polynomials; this flexible functional form enabled us to predict the probability of having a job start.[44]

Univariate and multivariable Poisson regression analyses were used to calculate incident rate ratios (IRR) and 95% CIs to examine the associations between job start (RTW) and individual, personal and external factors. The unadjusted models contained age and gender and the adjusted models contained all the variables in the study. Overall differences between the categories are reported and for ordered variables (length of unemployment, highest qualification, number of health conditions disclosed to advisor, SIMD quintiles), a linear trend across categories was also determined. Predicted probabilities of RTW were estimated using postestimation commands following regression modelling, with illustrative results shown for people aged 25 and 50 years when appropriate. All analyses were carried out using Stata V.14.

**Table 1** Descriptive characteristics and job start of the SOPIE cohort by benefit type

| | JSA clients | | ESA clients | |
|---|---|---|---|---|
| | **No. of clients** | **No. of clients with job start (% of each category with job start)** | **No. of clients** | **No. of clients with job start (% of each category with job start)** |
| | **(% of total JSA clients)** | | **(% of total ESA clients)** | |
| **Benefit type** | **8996** | **5612 (62.4%)** | **4322** | **867 (20.1%)** |
| **Individual factors** | | | | |
| Age (years) | | | | |
| <50 | 7590 (84.4%) | 4919 (64.8%) | 3000 (69.4%) | 685 (22.8%) |
| >50 | 1406 (15.6%) | 693 (49.3%) | 1322 (30.6%) | 182 (13.8%) |
| Gender | | | | |
| Male | 5799 (64.5%) | 3754 (64.7%) | 2260 (52.3%) | 450 (19.9%) |
| Female | 3197 (35.5%) | 1858 (58.1%) | 2062 (47.7%) | 417 (20.2%) |
| Length of prior unemployment | | | | |
| 0–6 months | 638 (7.1%) | 476 (74.6%) | 128 (3.0%) | 73 (57.0%) |
| 7–12 months | 2034 (22.6%) | 1498 (73.7%) | 264 (6.1%) | 118 (44.7%) |
| 1–2 years | 3510 (39.0%) | 2416 (68.3%) | 733 (17.0%) | 261 (35.6%) |
| 3–5 years | 1072 (11.9%) | 571 (53.3%) | 802 (18.6%) | 173 (21.6%) |
| 6–10 years | 886 (9.9%) | 399 (45.0%) | 885 (20.5%) | 119 (13.5%) |
| 11+ years | 856 (9.5%) | 252 (29.4%) | 1510 (34.9%) | 123 (8.2%) |
| Highest qualification | | | | |
| Degree or higher | 580 (6.5%) | 436 (75.2%) | 165 (3.8%) | 64 (38.8%) |
| A levels/NVQ level 3 and equivalent | 1443 (16.0%) | 1006 (69.7%) | 480 (11.1%) | 149 (31.0%) |
| Five or more GCSEs grades A*–C and equivalent | 1564 (17.4%) | 1094 (70.0%) | 468 (10.8%) | 134 (28.6%) |
| Under 5 GCSEs A*–C and equivalent | 2145 (23.8%) | 1317 (61.4%) | 975 (22.6%) | 185 (19.0%) |
| Below GSCE level | 3264 (36.3%) | 1759 (53.9%) | 2234 (51.7%) | 335 (15.0%) |
| Ethnicity | | | | |
| White British | 7950 (88.4%) | 4906 (61.7%) | 4062 (94.0%) | 813 (20.0%) |
| Other | 1046 (11.4%) | 706 (67.5%) | 260 (6.0%) | 54 (20.8%) |
| Have health concerns which believe will affect ability to work | | | | |
| No | 7247 (80.6%) | 4984 (68.8%) | 255 (5.9%) | 126 (49.4%) |
| Yes | 1749 (19.4%) | 628 (35.9%) | 4067 (94.1%) | 741 (18.2%) |
| Number of health conditions disclosed | | | | |
| 0 | 6365 (70.8%) | 4399 (69.1%) | | |
| 0 and 1 | | | 1290 (29.8%) | 381 (29.5%) |
| 1 | 1727 (19.2%) | 905 (52.4%) | | |
| 2 | 608 (6.8%) | 239 (39.3%) | 1396 (32.3%) | 296 (21.2%) |
| 3 | | | 896 (20.7%) | 123 (13.7%) |
| 3 or more | 296 (3.3%) | 69 (23.3%) | | |
| 4 | | | 425 (9.8%) | 40 (9.4%) |
| 5 or more | | | 315 (7.3%) | 27 (8.6%) |
| Client perception of job start—When do you see yourself starting work? | | | | |
| Within 1 month | 2188 (24.3%) | 1646 (75.2%) | 92 (2.1%) | 56 (60.9%) |
| 2–3 months | 3268 (36.3%) | 2258 (69.1%) | 248 (5.7%) | 142 (57.3%) |
| 4–6 months | 875 (9.7%) | 486 (55.5%) | 262 (6.1%) | 117 (44.7%) |
| >6 months | 397 (4.4%) | 115 (29.0%) | 1302 (30.1%) | 140 (10.8%) |
| Do not know | 2268 (25.2%) | 1107 (48.8%) | 2418 (56.0%) | 412 (17.0%) |

**Table 1** Continued

| Benefit type | JSA clients | | ESA clients | |
|---|---|---|---|---|
| | No. of clients | No. of clients with job start (% of each category with job start) | No. of clients | No. of clients with job start (% of each category with job start) |
| | (% of total JSA clients) | | (% of total ESA clients) | |
| | 8996 | 5612 (62.4%) | 4322 | 867 (20.1%) |
| **Personal circumstances** | | | | |
| Caring responsibility for anyone other than children | | | | |
| No | 8561 (95.2%) | 5398 (63.0%) | 4046 (93.6%) | 822 (20.3%) |
| Yes | 435 (4.8%) | 214 (49.2%) | 273 (6.4%) | 45 (16.3%) |
| Housing | | | | |
| Homeowner | 401 (4.5%) | 261 (65.1%) | 323 (7.5%) | 95 (29.4%) |
| Living with family | 3049 (33.9%) | 2155 (69.4%) | 609 (14.1%) | 146 (24.0%) |
| Rented private | 1304 (14.5%) | 850 (65.2%) | 563 (13.0%) | 126 (22.4%) |
| Rented social | 3853 (42.8%) | 2212 (57.4%) | 2655 (61.4%) | 469 (17.7%) |
| Insecure | 389 (4.3%) | 174 (44.7%) | 172 (4.0%) | 31 (18.0%) |
| Parental status | | | | |
| No children | 4892 (54.4%) | 3218 (65.8%) | 1693 (39.2%) | 329 (19.4%) |
| Children, two parent family | 622 (6.9%) | 406 (65.3%) | 288 (6.7%) | 89 (30.9%) |
| Children, shared custody/not living with you | 1254 (13.9%) | 735 (58.6%) | 711 (16.5%) | 145 (20.4%) |
| Children, lone parent family | 1369 (15.2%) | 800 (58.4%) | 597 (13.8%) | 135 (22.6%) |
| Children, adults living at home/adults not living at home | 859 (9.6%) | 453 (52.7) | 1033 (23.9%) | 169 (16.4%) |
| External factors | | | | |
| SIMD quintiles | | | | |
| 1 (most deprived) | 4779 (53.1%) | 2899 (60.7%) | 2362 (54.7%) | 411 (17.4%) |
| 2 | 2075 (23.1%) | 1302 (62.8%) | 1013 (23.4%) | 219 (21.6%) |
| 3 | 1151 (12.8%) | 707 (61.4%) | 550 (12.7%) | 130 (23.6%) |
| 4 | 603 (6.7%) | 419 (69.5%) | 226 (5.2%) | 57 (25.2%) |
| 5 (least deprived) | 388 (4.3%) | 285 (73.5%) | 171 (4.0%) | 50 (29.2%) |
| Sixfold urban rural classification | | | | |
| 1 Large urban areas | 4573 (50.8%) | 2891 (63.2%) | 2578 (59.7%) | 467 (18.1%) |
| 2 Other urban areas | 3176 (35.3%) | 1962 (61.8%) | 1237 (28.6%) | 275 (22.2%) |
| 3 Accessible small towns | 623 (6.9%) | 395 (63.4%) | 232 (5.4%) | 57 (24.6%) |
| 4 Remote small towns | 173 (1.9%) | 87 (50.3%) | 69 (1.6%) | 20 (29.0%) |
| 5 Accessible rural | 333 (3.7%) | 211 (63.4%) | 151 (3.5%) | 38 (25.2%) |
| 6 Remote rural | 118 (1.3%) | 66 (55.9%) | 55 (1.3%) | 10 (18.2%) |

Test of association on all variables and benefit type, p<0.001, except for test of association on SIMD quintiles and benefit type, p<0.05.
ESA, Employment and Support Allowance; GCSE, General Certificate of Secondary Education; JSA, Jobseeker's Allowance; NVQ, National Vocational Qualification; SIMD, Scottish Index of Multiple Deprivation; SOPIE, Supporting Older people into Employment.

## Patient/client involvement

No patients/clients were involved in developing the hypothesis, the specific aims or the research questions, nor were they involved in developing plans for design or implementation of the study. No clients were involved in the interpretation of study results or write up of the manuscript. Clients did provide feedback on emerging findings at yearly stakeholder meetings and a final study symposium.

## RESULTS
### Cohort demographics and job start

Of the SOPIE cohort of 13 318 clients, 8996 (68%) were claiming JSA and 4322 (32%) were claiming ESA. Table 1 shows the descriptive statistics of the cohort by benefit type for all variables included in the study. As expected, due to welfare benefit rules, more ESA clients were aged 50 years and over (ESA clients 31%; JSA clients 16%). There

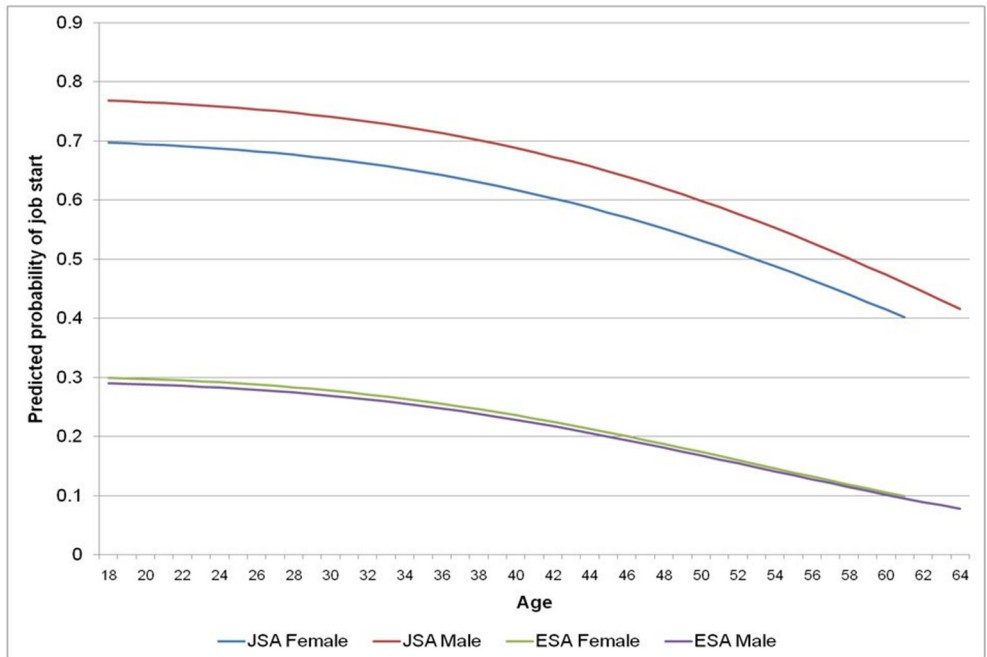

**Figure 1** Predicted probability of JSA and ESA clients having a job start by age during the 2-year Work Programme. ESA, Employment and Support Allowance; JSA, Jobseeker's Allowance.

were statistically significant differences (p<0.001) between the two client groups across all variables, except for the test of association between SIMD quintiles (p<0.05). The table also shows jobs starts for each variable, with 5612 JSA clients (62%) and 867 ESA clients (20%) having at least one job start during the 2-year programme.

### The influence of age and job start
Figure 1 shows how the predicted probability of JSA and ESA clients having a job start at any point during the 2-year WP varies according to baseline age and gender.

There is a strong negative relationship between age and having a job start for both JSA and ESA clients. Male JSA clients have a higher probability of job start compared with female JSA clients at all ages. However, there is no difference in the probability of a job start for male and female ESA clients.

### The influence of age and job start during the 2-year WP intervention
Figures 2 and 3 show the predicted probability of job start for all four 3-month periods in year 1 of the WP, and the

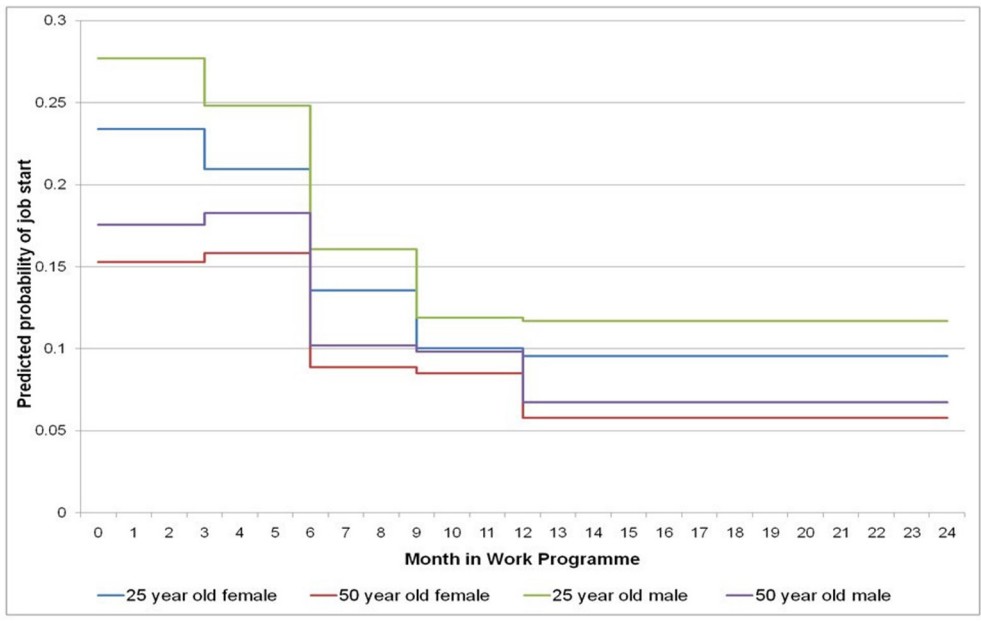

**Figure 2** Step graph showing predicted probability of female and male *JSA* clients aged 25 and 50 years having a job start in a 3-month period. JSA, Jobseeker's Allowance.

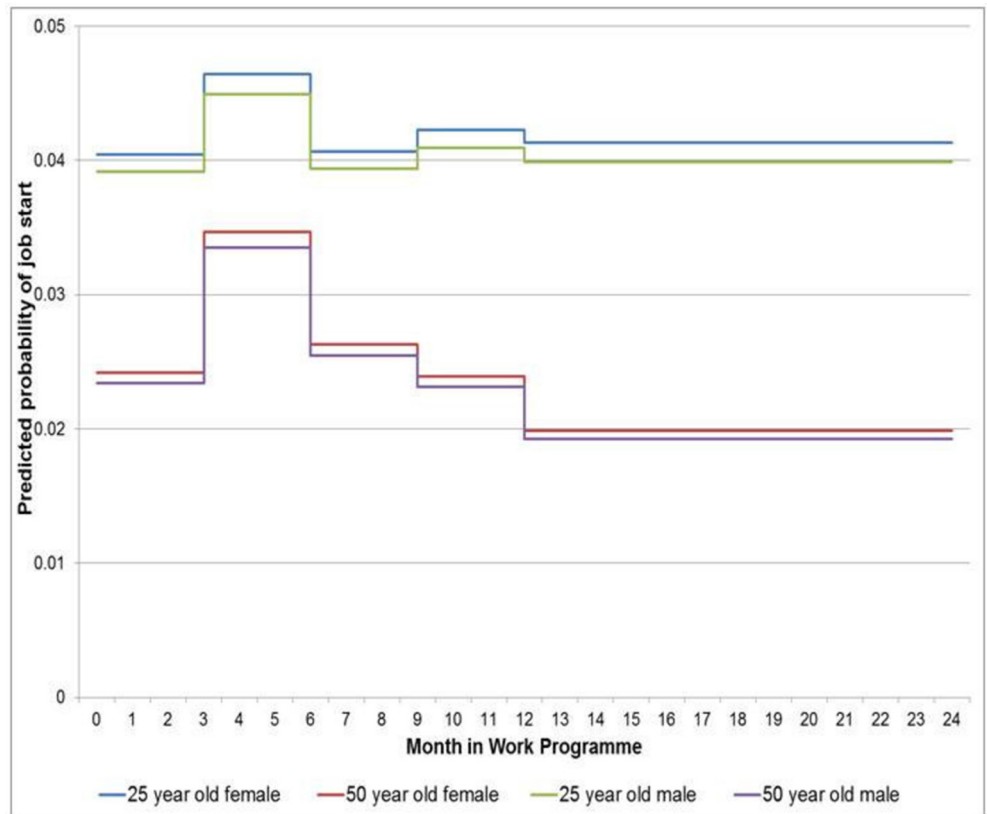

**Figure 3** Step graph showing predicted probability of female and male *ESA* clients aged 25 and 50 years having a job start in a 3-month period. ESA, Employment and Support Allowance.

average for all 3-month periods in the whole of year 2 of the WP, for the female and male JSA and ESA clients aged 25 and 50 years (Figures 2 and 3, respectively).

There are marked differences both by age and benefit type when clients are more likely to have a job start throughout the 2-year programme. Younger JSA clients have the highest predicted probability of job start in the first 3 months of their participation in the programme (females 23%, males 28%), which falls off every 3 months through the programme to 9.6% for females and 11.7% for males in an average 3-month period in year 2 of the programme. Older JSA clients have a similar probability of a job start by gender in the first 6 months of the programme (female 15%, male 18%), which decreases to 5.8% and 6.8% for females and males respectively in an average 3-month period in year 2 (figure 2).

ESA clients, with more health conditions, have a much lower probability of job start compared with JSA clients, particularly in the first 3 months of the programme (ESA clients aged 25 years 4%, 50 years 2.4%). Apart from a small increase (<1%) between 4 to 6 months the probability of a job start remains largely unchanged for ESA clients aged 25 and 50 years for the remainder of the programme, with the younger clients having a higher probability (figure 3).

### Factors associated with job start

Unadjusted and multivariable adjusted IRR for factors associated with having a job start for JSA clients are shown in table 2. Unadjusted analyses showed that female JSA clients were 16% less likely than male JSA clients to have a job start but this association was entirely attenuated when adjusting for all other factors. The length of unemployment before joining the WP was a strong predictor of job start. Clients having educational qualifications under five General Certificate of Secondary Education (GCSEs, or equivalent) or below GSCE level were 17% and 23%, respectively less likely to have a job start compared with those with a degree or higher. Unadjusted analyses showed that compared with white British clients, other clients were 12% more likely to have a job start (but when adjusted this association was not significant).

In terms of clients' health, disclosing one health condition was associated with a 12% decrease, two health conditions a 24% decrease and three or more health conditions a 45% decrease in the likelihood of having a job start compared with disclosing no health conditions. Having health concerns which the client believed would affect their ability to work versus having no health concerns, was associated with a 36% decrease in the likelihood of having a job start. Client perception of job start was a strong predictor of job start.

In terms of personal circumstances having caring responsibilities for anyone other than children and living in insecure housing (which included temporary housing and homelessness) were associated with a decreased likelihood of job start. Although unadjusted analyses showed

**Table 2** Factors associated with job start for *JSA clients*

| Variable | Unadjusted IRR (95% CI)* | P values | Adjusted IRR (95% CI)† | P values |
|---|---|---|---|---|
| **Individual factors** | | | | |
| Gender | | <0.001 | | 0.435 |
| Male | 1 | | 1 | |
| Female | 0.84 (0.80 to 0.89) | | 0.95 (0.89 to 1.02) | |
| Length of prior unemployment | | <0.001 | | <0.001 |
| 0–6 months | 1 | | 1 | |
| 7–12 months | 0.95 (0.85 to 1.05) | | 0.98 (0.88 to 1.09) | |
| 1–2 years | 0.80 (0.72 to 0.88) | | 0.84 (0.76 to 0.93) | |
| 3–5 years | 0.51 (0.45 to 0.58) | | 0.63 (0.56 to 0.72) | |
| 6–10 years | 0.39 (0.34 to 0.44) | | 0.51 (0.44 to 0.58) | |
| 11+ years | 0.26 (0.21 to 0.28) | | 0.39 (0.33 to 0.45) | |
| Highest qualification | | <0.001 | | <0.01 |
| Degree or higher | 1 | | 1 | |
| A levels/NVQ level 3 and equivalent | 0.83 (0.74 to 0.92) | | 0.95 (0.84 to 1.06) | |
| Five or more GCSEs grades A*–C and equivalent | 0.79 (0.71 to 0.89) | | 0.94 (0.84 to 1.06) | |
| Under 5 GCSEs A*–C and equivalent | 0.63 (0.56 to 0.70) | | 0.83 (0.74 to 0.93) | |
| Below GSCE level | 0.55 (0.49 to 0.61) | | 0.76 (0.68 to 0.85) | |
| Ethnicity | | <0.01 | | 0.476 |
| White British | 1 | | 1 | |
| Other | 1.12 (1.03 to 1.21) | | 0.96 (0.88 to 1.04) | |
| Have health concerns which believe will affect ability to work | | <0.001 | | <0.001 |
| No | 1 | | 1 | |
| Yes | 0.41 (0.37 to 0.44) | | 0.64 (0.58 to 0.70) | |
| Number of health conditions disclosed | | <0.001 | | <0.001 |
| 0 | 1 | | 1 | |
| 1 | 0.66 (0.61 to 0.71) | | 0.88 (0.82 to 0.95) | |
| 2 | 0.46 (0.40 to 0.52) | | 0.76 (0.66 to 0.88) | |
| 3 or more | 0.25 (0.20 to 0.32) | | 0.55 (0.43 to 0.71) | |
| Client perception of job start—When do you see yourself starting work? | | <0.001 | | <0.001 |
| Within 1 month | 1 | | 1 | |
| 2–3 months | 0.81 (0.76 to 0.87) | | 0.83 (0.78 to 0.89) | |
| 4–6 months | 0.55 (0.50 to 0.61) | | 0.65 (0.58 to 0.72) | |
| >6 months | 0.25 (0.21 to 0.31) | | 0.41 (0.34 to 0.50) | |
| Do not know | 0.49 (0.45 to 0.53) | | 0.64 (0.59 to 0.69) | |
| **Personal circumstances** | | | | |
| Caring responsibility for anyone other than children | | <0.001 | | <0.001 |
| No | 1 | | 1 | |
| Yes | 0.74 (0.64 to 0.85) | | 0.75 (0.66 to 0.86) | |
| Housing status | | <0.001 | | <0.001 |
| Homeowner | 1 | | 1 | |
| Living with family | 0.80 (0.69 to 0.92) | | 0.99 (0.86 to 1.14) | |
| Rented private | 0.80 (0.69 to 0.92) | | 1.00 (0.87 to 1.16) | |
| Rented social | 0.67 (0.59 to 0.76) | | 0.93 (0.81 to 1.07) | |
| Insecure | 0.41 (0.34 to 0.50) | | 0.65 (0.53 to 0.79) | |

Continued

**Table 2** Continued

| Variable | Unadjusted IRR (95% CI)* | P values | Adjusted IRR (95% CI)† | P values |
|---|---|---|---|---|
| **Individual factors** | | | | |
| Parental status | | <0.01 | | 0.505 |
| No children | 1 | | 1 | |
| Children, two parent family | 0.98 (0.88 to 1.09) | | 0.98 (0.87 to 1.09) | |
| Children, shared custody/not living with you | 0.83 (0.77 to 0.91) | | 0.93 (0.85 to 1.01) | |
| Children, lone parent family | 0.87 (0.79 to 0.96) | | 1.03 (0.94 to 1.14) | |
| Children, adults living at home/adults not living at home | 0.99 (0.88 to 1.11) | | 1.08 (0.96 to 1.21) | |
| External factors | | | | |
| SIMD quintiles | | <0.001 | | 0.145 |
| 1 (most deprived) | 1 | | 1 | |
| 2 | 1.08 (1.01 to 1.15) | | 1.03 (0.97 to 1.11) | |
| 3 | 1.06 (0.98 to 1.15) | | 1.00 (0.92 to 1.09) | |
| 4 | 1.26 (1.14 to 1.40) | | 1.10 (0.99 to 1.22) | |
| 5 (least deprived) | 1.42 (1.26 to 1.61) | | 1.12 (0.99 to 1.28) | |
| Sixfold urban rural classification | | <0.05 | | 0.069 |
| 1 Large urban areas | 1 | | 1 | |
| 2 Other urban areas | 0.96 (0.90 to 1.01) | | 0.94 (0.89 to 1.00) | |
| 3 Accessible small towns | 1.00 (0.90 to 1.11) | | 1.02 (0.91 to 1.13) | |
| 4 Remote small towns | 0.66 (0.54 to 0.82) | | 0.73 (0.59 to 0.91) | |
| 5 Accessible rural | 0.99 (0.86 to 1.14) | | 0.98 (0.85 to 1.13) | |
| 6 Remote rural | 0.82 (0.64 to 1.05) | | 0.76 (0.59 to 0.97) | |

Linear trends are shown for length of unemployment, highest qualification, number of health conditions and SIMD quintiles. All other variables show overall trend.
*Model contained age and gender.
†Model adjusted for age, gender, length of unemployment, highest qualification, ethnicity, health concerns, number of health conditions, client perception of job start, caring responsibility, housing status, parental status, SIMD quintiles and sixfold urban rural classification.
GCSE, General Certificate of Secondary Education; IRR, incident rate ratios; JSA, Jobseeker's Allowance; SIMD, Scottish Index of Multiple Deprivation.

significant associations with parental status, when adjusted all associations were lost. In terms of external factors, JSA clients living in SIMD quintile 1 were less likely to have a job start, and those living in large urban areas were more likely to have a job start; however, the associations were lost in the adjusted model.

Unadjusted and multivariable adjusted IRR for factors associated with having a job start for ESA clients are shown in table 3. Females were as likely as male clients to have a job start. Increasing length of unemployment was significantly associated with a lower probability of having a job start. While there was a relationship between education and RTW in the unadjusted model, this disappeared in the fully adjusted model. Non-white British clients were less likely to have a job start.

For ESA clients, disclosing health conditions was associated with a 16% decrease (two health conditions), 32% decrease (three health conditions), 51% decrease (four health conditions) and 49% decrease (five or more health conditions) in the likelihood of having a job start compared with disclosing zero or one health conditions. Having health concerns which the client believed would affect their ability to work versus having no health concerns, was associated with a 32% decrease in the likelihood of having a job start. Clients' perceptions of job start were a strong predictor of job start. Compared with those clients who thought they would start work within 1 month, clients who thought they would start work in >4 months were significantly associated with a reduction in job start in the adjusted model. In terms of personal circumstances, only parental status was significantly associated with job start, and those with no children less likely to have a job start. In the unadjusted model, those clients living in quintiles 2–5 were more likely to have a job start than clients living in quintile 1 (most deprived), but this association was lost after adjustment in the full model. ESA clients living in small towns and accessible rural areas were more likely to have a job start than those living in large urban areas; however, the effect was lost in the full model.

## DISCUSSION

This study sought to investigate the role of age, health and other factors associated with RTW among two client groups engaging with the 2-year WP. There is a strong

**Table 3** Factors associated with job start for *ESA clients*

| Variable | Unadjusted IRR (95% CI)* | P values | Adjusted IRR (95% CI)† | P values |
|---|---|---|---|---|
| **Individual factors** | | | | |
| Gender | | 0.632 | | 0.141 |
| Male | 1 | | 1 | |
| Female | 1.03 (0.90 to 1.18) | | 1.11 (0.95 to 1.29) | |
| **Length of prior unemployment** | | <0.001 | | <0.001 |
| 0–6 months | 1 | | 1 | |
| 7–12 months | 0.67 (0.50 to 0.90) | | 0.71 (0.53 to 0.96) | |
| 1–2 years | 0.49 (0.38 to 0.64) | | 0.57 (0.44 to 0.75) | |
| 3–5 years | 0.27 (0.20 to 0.35) | | 0.39 (0.29 to 0.51) | |
| 6–10 years | 0.16 (0.12 to 0.21) | | 0.25 (0.18 to 0.33) | |
| 11+ years | 0.10 (0.07 to 0.13) | | 0.18 (0.13 to 0.25) | |
| Highest qualification | | <0.001 | | 0.312 |
| Degree or higher | 1 | | 1 | |
| A levels/NVQ level 3 and equivalent | 0.78 (0.58 to 1.05) | | 1.16 (0.85 to 1.58) | |
| Five or more GCSEs grades A*–C and equivalent | 0.68 (0.51 to 0.92) | | 1.12 (0.82 to 1.54) | |
| Under 5 GCSEs A*–C and equivalent | 0.44 (0.33 to 0.59) | | 0.86 (0.63 to 1.17) | |
| Below GSCE level | 0.36 (0.27 to 0.47) | | 0.85 (0.64 to 1.14) | |
| **Ethnicity** | | 0.629 | | <0.05 |
| White British | 1 | | 1 | |
| Other | 1.07 (0.81 to 1.41) | | 0.71 (0.53 to 0.95) | |
| Have health concerns which believe will affect ability to work | | <0.000 | | <0.000 |
| No | 1 | | 1 | |
| Yes | 0.30 (0.25 to 0.37) | | 0.68 (0.55 to 0.84) <0.001 | |
| **Number of health conditions disclosed** | | <0.000 | | <0.000 |
| 0+1 (note 121 clients disclosed 0 HCs) | 1 | | 1 | |
| 2 | 0.68 (0.58 to 0.79) | | 0.84 (0.72 to 0.99) | |
| 3 | 0.44 (0.36 to 0.54) | | 0.67 (0.54 to 0.83) | |
| 4 | 0.30 (0.22 to 0.42) | | 0.49 (0.35 to 0.68) | |
| 5 or more | 0.28 (0.19 to 0.42) | | 0.52 (0.35 to 0.78) | |
| Client perception of job start—When do you see yourself starting work? | | <0.000 | | <0.000 |
| Within 1 month | 1 | | 1 | |
| 2–3 months | 0.81 (0.60 to 1.11) | | 0.97 (0.71 to 1.34) | |
| 4–6 months | 0.51 (0.37 to 0.71) | | 0.69 (0.49 to 0.97) | |
| >6 months | 0.11 (0.08 to 0.15) | | 0.24 (0.17 to 0.34) | |
| Do not know | 0.17 (0.13 to 0.22) | | 0.33 (0.24 to 0.45) | |
| **Personal circumstances** | | | | |
| Caring responsibility for anyone other than children | | 0.313 | | 0.575 |
| No | 1 | | 1 | |
| Yes | 0.86 (0.63 to 1.16) | | 0.89 (0.65 to 1.20) | |
| **Housing status** | | <0.001 | | 0.066 |
| Homeowner | 1 | | 1 | |
| Living with family | 0.51 (0.38 to 0.67) | | 0.68 (0.50 to 0.90) | |
| Rented private | 0.54 (0.41 to 0.71) | | 0.71 (0.53 to 0.93) | |
| Rented social | 0.45 (0.36 to 0.56) | | 0.73 (0.58 to 0.93) | |
| Insecure | 0.38 (0.25 to 0.58) | | 0.61 (0.40 to 0.93) | |

Continued

**Table 3** Continued

| Variable | Unadjusted IRR (95% CI)* | P values | Adjusted IRR (95% CI)† | P values |
|---|---|---|---|---|
| Parental status | | 0.351 | | <0.01 |
| No children | 1 | | 1 | |
| Children, two parent family | 1.61 (1.27 to 2.04) | | 1.41 (1.10 to 1.80) | |
| Children, shared custody/not living with you | 1.06 (0.87 to 1.29) | | 1.18 (0.97 to 1.45) | |
| Children, lone parent family | 1.12 (0.90 to 1.39) | | 1.25 (0.99 to 1.56) | |
| Children, adults living at home/adults not living at home | 1.13 (0.92 to 1.40) | | 1.26 (1.02 to 1.57) | |
| **External factors** | | | | |
| SIMD quintiles | | <0.001 | | 0.331 |
| 1 (most deprived) | 1 | | 1 | |
| 2 | 1.28 (1.09 to 1.51) | | 1.09 (0.92 to 1.29) | |
| 3 | 1.42 (1.17 to 1.75) | | 0.97 (0.78 to 1.19) | |
| 4 | 1.58 (1.20 to 2.09) | | 1.21 (0.91 to 1.61) | |
| 5 (least deprived) | 1.86 (1.39 to 2.50) | | 1.17 (0.86 to 1.59) | |
| Sixfold urban rural classification | | <0.01 | | 0.163 |
| 1 Large urban areas | 1 | | 1 | |
| 2 Other urban areas | 1.22 (1.05 to 1.42) | | 1.04 (0.89 to 1.21) | |
| 3 Accessible small towns | 1.39 (1.06 to 1.83) | | 1.15 (0.87 to 1.53) | |
| 4 Remote small towns | 1.64 (1.05 to 2.56) | | 1.56 (0.98 to 2.46) | |
| 5 Accessible rural | 1.51 (1.08 to 2.10) | | 1.08 (0.77 to 1.51) | |
| 6 Remote rural | 1.06 (0.57 to 1.98) | | 0.74 (0.39 to 1.41) | |

Linear trends are shown for length of unemployment, highest qualification, number of health conditions and SIMD quintiles. All other variables show overall trend.

*Model contained age and gender.

†Model adjusted for age, gender, length of unemployment, highest qualification, ethnicity, health concerns, number of health conditions, client perception of job start, caring responsibility, housing status, parental status, SIMD quintiles and sixfold urban rural classification.

ESA, Employment and Support Allowance; GCSE, General Certificate of Secondary Education; HC, health conditions; IRR, incident rate ratios; SIMD, Scottish Index of Multiple Deprivation.

continuous negative relationship between age and having a job start for both JSA and ESA clients, with no clear evidence for a specific age at which RTW becomes much less likely. For all JSA clients, the probability of a job start is highest in the first 3 months from joining the programme, decreasing progressively throughout rest of the programme. For ESA clients, the probability of a job start changes little throughout the WP. We further identified a range of factors which were associated with JSA and ESA clients having a job start, including some that have not been explored in previous RTW programmes. Our findings reveal some benefit type similarities as well as some interesting differences with the literature.

Reducing the disability employment gap and enabling more older people to work for longer are key policy challenges.[12 35] This study is therefore particularly important as it extends our understanding of the factors, including age and health as well as certain socioeconomic factors, which are associated with RTW and by investigating the JSA and ESA clients separately we have detected further differences which other studies were not able to discern.[8 37] The unique and rich SOPIE cohort also benefits from a 2-year follow-up on all clients. While several studies have reviewed RTW in welfare-to-work initiatives,

these have been limited in several ways, for example, they examine fewer explanatory variables and discrete age categories.[8 45 46] Thus, the size of the SOPIE cohort and the range of the variables collected are considerable strengths. While the availability of such rich data on RTW is a major strength of this study, there are also some limitations to the data. The research team only had access to the variables routinely collected and could not specify the data collection. Apart from the employment outcomes, the baseline data were generally collected when a client was first referred to the programme and data on external factors such as job opportunities is limited. While this study is limited to clients in Scotland, the results are generalisable to the rest of the UK.

In terms of the first research question, older age has been shown to be associated with a lower likelihood of RTW, consistent with many other studies.[8 45–49] However, figures 2 and 3 suggest that while in the first 9 months younger JSA clients have a higher likelihood of RTW, after that period the rates differ little by age. For ESA clients, the age differentials remain for the full 2 years of the programme. For research question 2, the highest predicted probability of job start is in the first 3 months of the programme for JSA clients. These clients could simply

be the most employable clients (eg, in terms of skills, experience, etc) and so be an example of 'creaming' (where the support provider prioritises those unemployed claimants with fewer barriers to work and who are therefore felt to be easier and more likely to move into paid work) as suggested in other studies.[50 51] In contrast although much lower than JSA clients, the highest predicted probability of job start for the ESA clients (both young and old) was between 4 and 6 months. This is unsurprising given the ESA clients had been unemployed for much longer than JSA clients, had more health conditions (table 1) and may require more support or time for the modification of health barriers or their perceived redundancy of skills. Both client groups have a decrease in RTW probability after 6 months indicating that increased support or a change in the type of support (including perhaps new approaches) may be required as time in the programme increases, or they may be seen as being so far from the labour market employers may not be willing to make sufficient workplace adjustments or feel financial incentives, such as wage subsidies, are needed[52] or consider them as being virtually unemployable in current labour market conditions.[53]

Research question 3 asked what other factors are associated with RTW for JSA and ESA clients. In addition to age, the factors we investigated were drawn from a broad framework of employability covering three main interrelated components, or sets of factors, which influence a person's employability: individual factors, personal circumstances and external factors.[54 55] Although previous studies have shown gender differences in the likelihood of RTW,[8 45–47 56] the gender difference that was observed for JSA clients is lost when all other factors were included in the adjusted model. Those with poorer employment records (especially longer periods of unemployment) on entering the WP were less likely to have a job start and this confirms much prior evidence of having recent work experience prior to claim/entry to programme for RTW.[8 45 46] Our results further confirm earlier evidence that higher qualifications are important in influencing RTW,[46 49] but interestingly not for ESA clients. While there was a relationship between educational qualifications and RTW in the unadjusted model, this disappeared in the fully adjusted model. Further investigation showed this to be due to the relationship between RTW and both length of unemployment and the client's perception of when they saw themselves starting a new job. Ethnicity was important for ESA clients (at the 5% significance level, although not significant for JSA clients), suggesting clients other than white British may require more appropriate support.

Health is a major obstacle to the re-employment of benefit claimants.[7 8 45 46] With increasing age, the prevalence of long-term conditions and disability also increases.[57–59] Interestingly, we found that 29% of the JSA clients disclosed at least one health condition (50% of the over 50 JSA clients), which would suggest that health is still a potential barrier to RTW for JSA clients and should not be ignored. As with other studies,

disclosing a health condition (and the novel finding of increasing number) was significantly associated with the decreased likelihood of a job start.[8 47 48 59] Furthermore, health beliefs were important for both client groups confirming the impact of psychosocial factors. Client perception of their likely job start was important for both client groups and negative expectations may reduce resilience and lead to self-fulfilling outcomes of lower RTW. This is consistent with other studies that have shown that clients' own assessment of their ability to RTW was a strong predictor.[60–62]

Personal circumstances included a range related to individuals' social and household circumstances. These may affect the ability, willingness or social pressure for someone to take up an employment opportunity.[63] For JSA clients, childcare had relatively few effects. Hence the, usually gendered, effects of having dependent children seem limited. However, for ESA clients having parental responsibilities actually increased the likelihood of a job start, perhaps due partly to psychosocial factors or a reflection of their level of disability. For non-child caring responsibilities there was a significant impact for JSA clients but the numbers were small (only 4.8% of JSA clients). Housing status was important for JSA clients, although in the adjusted analyses it would appear that it was the clients living in insecure housing that was driving much of this overall negative association. For those in temporary or sheltered housing, it is difficult to find a job as some employers may prefer those with a more fixed abode or it may be difficult for clients to actually apply of a job if they have no permanent address.[64]

In terms of external factors, as with a WP evaluation,[8] the unadjusted results show that areas of greatest deprivation were associated with lower RTW, but this association was lost in the full models for both client groups. In the unadjusted models, those JSA clients living in areas other than large urban areas were less likely to have a job start whereas ESA clients living in small towns and accessible rural areas were more likely to have a job start; however, these effects disappeared after adjusting for other factors.

It is widely recognised that being employed can improve a person's health and well-being and help reduce health inequalities.[2 5] The key findings of this study have important implications for policy makers. While the disability employment gap has been recognised in Government policy,[36] there is little evidence that current programmes will reduce barriers to the employment of ageing workers. While they will provide specialised support for those unemployed for over 2 years, our findings would suggest much earlier intervention is needed. This is also supported by evidence that the longer an individual is absent from work, the less likely it is that they will return, and early intervention for those off work sick has been shown to be effective.[65 66] Programmes for those with health conditions or disabilities are likely to be voluntary, but therefore may not engage individuals who, because of their unemployment, are more likely to have low mood, have an inappropriately pessimistic outlook, be socially

isolated and reluctant to access support which needs to be based on the biopsychosocial model.[67 68] Perceptions about ability to work are important, but these may have been influenced by health professionals or other advisors with little knowledge of occupational health, workplaces or access to vocational rehabilitation expertise.[65] This study clearly shows that for the individual there is an inverse relationship between job start and the number of health conditions highlighting the need for healthcare providers to include vocational rehabilitation as part of treatment pathways.[69] Linked to improving workability are education and retraining of ageing workers with medical conditions who may be unfit for their usual role and be disadvantaged because of poor IT or other skills.[15] At present in the UK, much of the educational focus has been on the young unemployed,[70] and new programmes need to include training to update and develop new skills for older workers.

While it is generally accepted that most work is beneficial to health,[5] the potential health impacts of engaging with the WP requires further evaluation and linkage of this cohort to National Health Service Scotland Information Services Division health data is planned. More nuanced estimates of contextual factors such as personal circumstances (including the influence of others in the household) and external factors (such as types of local labour demand, employer behaviours and transport provision) would be useful in refining their influence and importance. Future research also needs to evaluate the long-term vocational outcomes of RTW programmes and whether expected health benefits of RTW are realised by these programmes, particularly when distinguishing the types of jobs people enter and the possible increase in job precariousness and insecurity.

## CONCLUSION

Age, health and a variety of socioeconomic factors play an important role in influencing RTW for unemployed people and for people who have an illness, health condition or disability that makes it difficult to RTW. Other countries with similar types of programmes, supporting both disabled and other job seekers, may also find similar relationships between individual characteristics and personal circumstances of participants. The results from this study will help inform interventions focussing on addressing age-specific, health and biopsychosocial barriers for future RTW programmes with the aim of improving employment outcomes, so that individuals, society, employers and the wider economy can benefit from extending working lives.

**Author affiliations**
¹Healthy Working Lives Group, Institute of Health and Wellbeing, College of Medical, Veterinary and Life Sciences, University of Glasgow, Glasgow, UK
²MRC/CSO Social and Public Health Sciences Unit, Institute of Health and Wellbeing, College of Medical, Veterinary and Life Sciences, University of Glasgow, Glasgow, UK
³Management, Work and Organisation, University of Stirling, Stirling, UK
⁴Usher Institute of Population Health Sciences and Informatics, University of Edinburgh, Edinburgh, UK

**Acknowledgements** The authors would like to acknowledge the help of Paul de Pellette, Luke Jeavons, Cem Zobu at Ingeus, Oarabile Molaodi, Daniel Mackay at the University of Glasgow and the DWP.

**Contributors** EBM, JB, SVK, AHL, RWM and JF had the original idea and developed the overall research design. JB conducted the statistical analysis with significant advice from SVK, AHL, RWM and JF. JB wrote the first draft. All authors subsequently contributed and commented to the manuscript and approved the final version.

**Funding** This project (and researcher JB) is funded by the MRC as part of MRC LLHW Extending Working Lives Partnership Awards (MR/L006367/1). AHL and SVK are funded by the MRC (MC_UU_12017/13 and MC_UU_12017/15) and the Scottish Government Chief Scientist Office (SPHSU13 and SPHSU15). SVK is supported by a Fellowship from the Scottish Government Chief Scientist Office (SCAF/15/02). JF is funded by the MRC (MR/K023209/1) and Scottish Government Chief Scientist Office (SCPHRP), as well as the Usher Institute of Population Health Sciences and Informatics at the University of Edinburgh.

**Competing interests** None declared.

**Patient consent** Not required.

**Ethics approval** Ethical approval for this project was received from the University of Glasgow College of Social Sciences Research Ethics Committee (400140186).

**Provenance and peer review** Not commissioned; externally peer reviewed.

**Data sharing statement** Ingeus acts as data processor on behalf of the data controller, the DWP. Ingeus is able to share data with the University of Glasgow (UoG) as detailed in the MRC Collaboration Agreement and the Research Data Licence between UoG and DWP.

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
