## [Reviewer comments · BMJ Open]

ARTICLE DETAILS

TITLE (PROVISIONAL)	Age, health and other factors associated with return to work for those engaging with a welfare-to-work initiative: A cohort study of administrative data from the UK's Work Programme
AUTHORS	Brown, Judith; Katikireddi, Srinivasa; Leyland, Alastair; McQuaid, Ronald; Frank, John; Macdonald, Ewan

VERSION 1 – REVIEW

REVIEWER	Alex Collie Monash University, Australia
REVIEW RETURNED	06-Jul-2018

GENERAL COMMENTS	This is an elegant study of an important social program that sought to engage unemployed people in work. Evaluations of large social policy programs as this are critical to inform future policy. The authors describe a well designed and well conducted study. The major area for improvement is to consider how the findings of this study may be important for an international audience. Other nations such as my own (Australia) have enacted labour market activation programs that are similar to the JSA and ESA described, and thus the findings have potential significance beyond the UK. I would encourage the authors to at least describe how the work program approach compares (or not) to employment/welfare programs in other similar countries and what the implications of this study may be for those other countries. Minor comments: - Did the authors test for associations between predictors and outcomes in univariate analyses before entering all the variables in the adjusted model? If not, how did they select the predictors for inclusion?- In some places the manuscript assumes knowledge of UK policy environment which should not be assumed for an international audience. For example page 27 line 53 states that "Future programmes for those with health conditions or disabilities will be voluntary, and therefore....." I can only assume the authors are referring to a policy decision made by the UK government that I (and I suspect many other readers) am not aware of?- The conclusion section of the abstract refers to findings that are not presented in the results section of the abstract. Suggest a revision of the abstract to ensure it flows appropriately.- There seems to be opportunity to aggregate some of the predictor categories to present a simpler regression model, to ease interpretation of findings, and to account for small sample
---

	sizes in come categories. For example the SIMD deciles could be presented as quintiles? The urban rural classifications could be aggregated into a smaller number of categories? - There are multiple spelling or typographical errors throughout the manuscript (e.g., page 19 line 3 spelling of 'virtually').
--	--

REVIEWER	Hung-Yi Chuang Kaohsiung Medical University, Taiwan
REVIEW RETURNED	12-Jul-2018

GENERAL COMMENTS	This study to explored the role of individual age, health and personal circumstances in returning to work (RTW), using Employment Support Allowance (ESA) and Jobseeker's Allowance (JSA) data with piecewise Poisson regression methods used to calculate incident rate ratios (IRR). They found JSA and ESA were showing important differences in return to work. JSA clients (62%) were more likely to return to work than ESA clients (20%). Furthermore, age was an important role in influencing return to work, however the length of unemployment, training and education, the management of multimorbidity, and the individual's perception of the likelihood of job start, which made it difficult to RTW. The manuscript has been well organized to publish with minor revision:  1. Age of JSA was younger than ESA clients in system nature, please offer more discussion. 2. The proportions of RTW in JSA and ESA were more than 62% and 20% respectively; thus why Poisson regressions were used? Why not logistic regression? Please offer more description of piecewise Poisson regression method.
--

VERSION 1 – AUTHOR RESPONSE

Reviewer(s)' Comments to Author:

Reviewer: 1

Reviewer Name: Alex Collie

Institution and Country: Monash University, Australia

Please state any competing interests or state 'None declared': none declared

Please leave your comments for the authors below This is an elegant study of an important social program that sought to engage unemployed people in work. Evaluations of large social policy programs as this are critical to inform future policy. The authors describe a well designed and well conducted study. The major area for improvement is to consider how the findings of this study may be important for an international audience. Other nations such as my own (Australia) have enacted labour market activation programs that are similar to the JSA and ESA described, and thus the findings have potential significance beyond the UK. I would encourage the authors to at least describe

how the work program approach compares (or not) to employment/welfare programs in other similar countries and what the implications of this study may be for those other countries.

2. Author Response: We have added new text on how the Work Programme approach compares to employment/welfare programs in other similar countries and included the term active labour market policies – ‘The WP was the UK Government’s flagship welfare-to-work initiative to help those more detached from the labour market to enter employment and reduce the time people spent on benefits. The design of the WP has parallels with other Organisation for Economic Cooperation and Development (OECD) countries active labour market policies for those on welfare or unemployed, in terms of moves towards delivery of general and specialist employment services through networks of private and not-for-profit organisations, usually through employment outcomes based performance contracts, with a variety of forms of procurement.[24 25] In addition to the WP, outsourcing included services for the disabled in countries such as Australia and the Netherlands (mainly to not-for-profit or private organisations), Sweden, Denmark and USA.[26-30]’

We have included six new international references (from Australia, Netherlands, Sweden, Denmark and the USA) in the introduction (p6) and text on implications for other countries in the conclusion (p24). We have also added two further relevant European references in the conclusion.

Minor comments:

- Did the authors test for associations between predictors and outcomes in univariate analyses before entering all the variables in the adjusted model? If not, how did they select the predictors for inclusion?

3. Author Response: Yes, we did test for associations between predictors and outcomes in univariate analyses (column 2 in Tables 2 & 3) before entering all the variables in the adjusted model (column 4 in Tables 2 & 3). In order to make this clearer we amended the text on page 12 to ‘Univariate and multivariable Poisson regression analyses were used to calculate incident rate ratios (IRR) and 95% confidence intervals (95% CI) to examine the associations between job start (RTW) and individual, personal and external factors.’

- In some places the manuscript assumes knowledge of UK policy environment which should not be assumed for an international audience. For example page 27 line 53 states that "Future programmes for those with health conditions or disabilities will be voluntary, and therefore....." I can only assume the authors are referring to a policy decision made by the UK government that I (and I suspect many other readers) am not aware of?

4. Author Response: We have updated this sentence (now on page 23) to ‘Programmes for those with health conditions or disabilities are likely to be voluntary, but therefore may not engage individuals who, because of their unemployment, are more likely to have low mood, have an inappropriately pessimistic outlook, be socially isolated and reluctant to access support which needs to be based on the biopsychosocial model.’

- The conclusion section of the abstract refers to findings that are not presented in the results section of the abstract. Suggest a revision of the abstract to ensure it flows appropriately.

5. Author Response: Thank you for pointing this out. We have updated the results and conclusion of the abstract and hope it now flows appropriately.

- There seems to be opportunity to aggregate some of the predictor categories to present a simpler regression model, to ease interpretation of findings, and to account for small sample sizes in some categories. For example the SIMD deciles could be presented as quintiles? The urban rural classifications could be aggregated into a smaller number of categories?

6. Author Response: As suggested by the reviewer we re-ran the unadjusted and adjusted models with SIMD quintiles and two aggregates of the urban rural classification (3-fold and 6-fold) to present a simpler regression model. We have now presented the SIMD data as quintiles and the urban rural classifications as the 6-fold classification. Results are similar except for one change: In the full model JSA clients living in large urban areas were as likely to have a job start as those clients not living in large urban areas ($p=0.069$)

Further when we were checking the models we found we had reported an error in one of the p values in the unadjusted ESA clients model (8 fold urban rural classification should have been $p<0.01$ not $p=0.678$. The p value for the unadjusted 6-fold urban rural classification is $p<0.01$ and this is shown in Table 3 on p38). The text has been changed accordingly on p17 ('ESA clients living in small towns and accessible rural areas were more likely to have a job start than those living in large urban areas, however the effect was lost in the full model'). As before any effect of urban rural classification is lost in the full model.

- There are multiple spelling or typographical errors throughout the manuscript (e.g., page 19 line 3 spelling of 'virtually').

7. Author Response: We checked and updated typographical errors throughout the manuscript and these are shown in the track changes.

Reviewer: 2

Reviewer Name: Hung-Yi Chuang

Institution and Country: Kaohsiung Medical University, Taiwan

Please state any competing interests or state 'None declared': None

Please leave your comments for the authors below This study to explored the role of individual age, health and personal circumstances in returning to work (RTW), using Employment Support Allowance (ESA) and Jobseeker's Allowance (JSA) data with piecewise Poisson regression methods used to calculate incident rate ratios (IRR). They found JSA and ESA were showing important differences in return to work. JSA clients (62%) were more likely to return to work than ESA clients (20%). Furthermore, age was an important role in influencing return to work, however the length of unemployment, training and education, the management of multimorbidity, and the individual's perception of the likelihood of job start, which made it difficult to RTW. The manuscript has been well organized to publish with minor revision:

1. Age of JSA was younger than ESA clients in system nature, please offer more discussion.

8. Author Response: The age of the JSA clients were younger than the ESA clients (median age: JSA clients 32; ESA clients 44). This is because 18 to 24 year old JSA clients were a priority group for the Work Programme. Also ESA clients were not separated by age and often people become disabled or experienced worsening health later in life. The text has been updated to explain this on pages 7 and 8 - 'Those [JSA clients] aged 18-24 years were a priority group and had to have been unemployed a

shorter period (usually 6 months) before entering the WP than most older groups.’ ‘ESA clients are not separated by age and often people become disabled or experience worsening health, and so join ESA, later in life. So generally ESA clients in our study are older than JSA clients.’

In addition we have added the following sentence to the results on p14:

‘As expected, due to welfare benefit rules, more ESA clients were aged 50 and over (ESA clients 31%; JSA clients 16%).’

2. The proportions of RTW in JSA and ESA were more than 62% and 20% respectively; thus why Poisson regressions were used? Why not logistic regression? Please offer more description of piecewise Poisson regression method.

9. Author Response: We have explained why Cox’s proportional hazards models were not appropriate and why we used piecewise Poisson regression. We could not use logistic regression as we wanted to include time to job start/return to work. We have offered more description of the piecewise Poisson regression method and modified the text in the methods section as follows:

‘To address the three research questions a mixture of descriptive statistics and regression analyses were used. All analyses were stratified by benefit type (JSA and ESA clients) given the large differences in RTW between the two groups. Counts and percentages were used to summarise categorical variables. The associations between benefit type and all the study variables were analysed using chi-squared tests. Cox’s proportional hazards models were initially used to determine the hazard ratios of clients returning to work but the proportional hazards assumption was not met. We therefore approximated the survival model using a piecewise Poisson regression model – equivalent to a Cox model with baseline hazard able to vary between sections.[43] Split times used in the models were as follows: 0 to 3 months, 3 to 6 months, 6 to 9 months, 9 to 12 months and 12 to 24 months (due to sample size a three month average probability is shown for the 12 to 24 month time period). We modelled age as a continuous variable using fractional polynomials; this flexible functional form enabled us to predict the probability of having a job start.[44]’

VERSION 2 – REVIEW

REVIEWER	Hung-Yi Chuang Kaohsiung Medical University, Taiwan
REVIEW RETURNED	15-Sep-2018
GENERAL COMMENTS	I have no other comments because the authors have revised the manuscript according to my previous comments.